# Chain-of-Thought Predictive Control

Zhiwei Jia [1]  Fangchen Liu [2]  Vineet Thumuluri [1]  Linghao Chen [3]  Zhiao Huang [1]  Hao Su [1]

## Abstract

We study generalizable policy learning from demonstrations for complex low-level control tasks (e.g., contact-rich object manipulations). We propose an imitation learning method that incorporates the idea of temporal abstraction and the planning capabilities from Hierarchical RL (HRL) in a novel and effective manner. As a step towards decision foundation models, our design can utilize scalable, albeit highly sub-optimal, demonstrations. Specifically, we find certain short subsequences of the demos, i.e. the chain-of-thought (CoT), reflect their hierarchical structures by marking the completion of subgoals in the tasks. Our model learns to dynamically predict the entire CoT as coherent and structured long-term action guidance and consistently outperforms typical two-stage subgoal-conditioned policies. On the other hand, such CoT facilitates generalizable policy learning as they exemplify the decision patterns shared among demos (even those with heavy noises and randomness). Our method, Chain-of-Thought Predictive Control (CoTPC), significantly outperforms existing ones on challenging low-level manipulation tasks from scalable yet highly sub-optimal demos.

## 1. Introduction

Hierarchical RL (HRL) (Hutsebaut-Buysse et al., 2022) has attracted much attention in the AI community as a promising direction for sample-efficient and generalizable policy learning. HRL tackles complex sequential decision-making problems by decomposing them into simpler and smaller sub-problems via temporal abstractions. In addition, many adopt a two-stage policy and possess the planning capabilities for high-level actions (i.e., subgoals or options) to achieve generalizable decision making. On the other hand, imitation learning (IL) remains one of the most powerful

---
[1]UC San Diego [2]UC Berkeley [3]Zhejiang University. Correspondence to: Zhiwei Jia <zjia@eng.ucsd.edu>.

Our code is available at this repo. Our project page is at this link.

approaches to training autonomous agents that interact with the environment digitally or physically. Without densely labelled rewards or on-policy / online interactions, IL usually casts policy learning as (self-)supervised learning with the potential to leverage large-scale pre-collected demonstrations. Given the recent success of large language models (LLMs), the promise of foundation models for decision making (Yang et al., 2023) seems to be significant, once internet-scale demonstration data become available and their sub-optimality is handled well. In this paper, we study how to perform generalizable IL with scalable yet sub-optimal demos by embracing the spirit of HRL in a novel way.

Despite the recent progress (Chen et al., 2021; Florence et al., 2022; Shafiullah et al., 2022; Liu et al., 2022; Ajay et al., 2022), it remains extremely challenging to solve low-level control tasks such as contact-rich object manipulations by IL in a scalable manner. Machine-generated demonstrations (Shridhar et al., 2020; Fishman et al., 2023; Gu et al., 2023) recently gain a lot of attention as they are scalably collected by either RL agents or heuristics / sampling-based planners. However, for (continuous) low-level control tasks, these demos are inherently highly sub-optimal for IL because of the underlying contact dynamics (Pfrommer et al., 2021) and the way they are generated. Their undesirable properties, namely being non-Markovian, highly noisy, discontinuous, and random, pose great challenges in both the optimization and the generalization of the IL policies. In particular, the noise & the discontinuity make the agent vulnerable to compounding errors (Ross & Bagnell, 2010), and the randomness suggests the common patterns among different trajectories can be unclear, hard to model and not generalizable. See detailed discussion in Sec. 4.1.

We find that, by adopting temporal abstraction and high-level planning in a novel way for IL, we can essentially enjoy the large-scale (albeit sub-optimal) demonstrations to achieve a significant performance boost on challenging tasks. On one hand, we leverage the hierarchical structures inherently presented in many low-level control tasks, especially object manipulations. We observe that certain subsequences of the demo trajectories (i.e., some key frames or key states) naturally mark the completion of their subgoals. For example, to accomplish a peg insertion task, in each trajectory, an agent has to first reach a state when it just grasps the peg, then another state when the angle of the

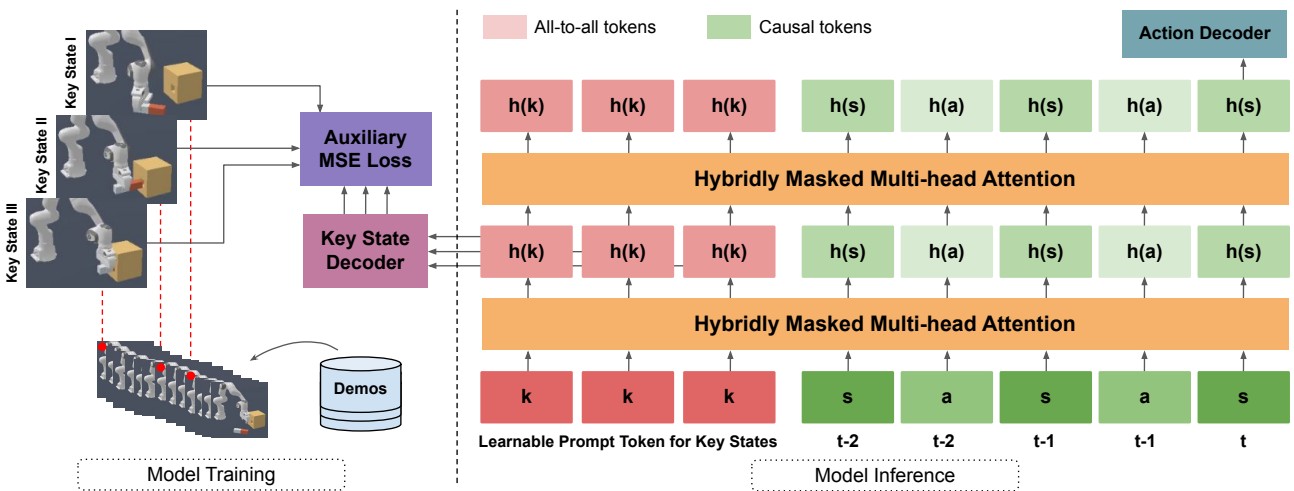

*Figure 1.* During training, CoTPC learns to jointly predict key states (the CoT) in the latent space and the current action with an MSE loss and a BC loss (omitted here). During inference, it predicts the entire CoT and the current action at each step, resembling closed-loop control. The prompt tokens for key states are all-to-all (can see any tokens up to the current step). The state and action tokens are causal (can only see previous tokens and the key state tokens). Only 2 attention layers and 3 timesteps in the history are shown for better display.

peg is just aligned with the hole, and finally a state when the peg is firmly pushed into the hole. These key states can be extracted by simple rules shared among different demo trajectories, which can be easily implemented in simulation or potentially automated with pre-trained vision-language models (zero-shot image retrieval). On the other hand, we apply high-level planning by learning joint predictions of key states and actions in a customized Transformer where both are further conditioned on the trajectory history. We call the key state sequences the chain-of-thought (CoT) (Wei et al., 2022) as they capture the multi-step nature of the tasks. At each inference step, our model updates the CoT predicted in the latent space and generates reliable actions accordingly. Note that common subgoal-conditioned policies in HRL only predict the immediate next subgoal once in a while, whereas ours is tailored for the IL setup and *dynamically predicts the entire CoT*. We demonstrate the advantages of dynamic, coherent and long-term subgoal planning and that the CoT predictions can be easily learned from demonstrations.

We call our method Chain-of-Thought Predictive Control (CoTPC). From an optimization perspective, it better leverages scalable yet sub-optimal demonstrations by utilizing hierarchical patterns shared even among demos with heavy noise and randomness. From a generalization perspective, CoTPC uses the impressive reasoning capabilities of Transformers (Brown et al., 2020) to achieve generalizable decision-making with predicted long-term high-level plans. We evaluate CoTPC on several challenging low-level object manipulation tasks, justify its design with extensive ablation studies, and further demonstrate its performance in a real-world robot setup. CoTPC outperforms several

strong baselines by a significant margin, especially for the peg insertion task where existing methods struggle.

## 2. Related Work

**Learning from Demonstrations (LfD)** Learning interactive agents from pre-collected demonstrations has been popular for policy learning due to its effectiveness and scalability. Roughly speaking, there are three categories: offline RL, online RL with auxiliary demos, and behavior cloning (BC). While offline RL approaches (Kumar et al., 2019; Fu et al., 2020; Levine et al., 2020; Kumar et al., 2020; Kostrikov et al., 2021; Chen et al., 2021; Wang et al., 2022) usually require demonstration with densely labelled rewards and methods that augment online RL with demos (Hester et al., 2018; Kang et al., 2018; Ross et al., 2011; Nair et al., 2020; Rajeswaran et al., 2017; Ho & Ermon, 2016; Pertsch et al., 2021; Singh et al., 2020) rely on on-policy interactions, BC (Pomerleau, 1988) formulates fully supervised or self-supervised learning problems. With its simplicity, zero sample complexity, and better practicality, BC has been extensively used in real-world environments, especially in robotics (Zeng et al., 2021; Florence et al., 2022; Qin et al., 2022; Zhang et al., 2018; Brohan et al., 2022; Rahmatizadeh et al., 2018; Florence et al., 2019; Zeng et al., 2020).

**Challenges in BC** Despite the wide applications of BC, a well-known shortcoming is the compounding error (Ross et al., 2011). One source of this is the distribution shift (i.e., covariate shift) between the demo data and the ones unrolled at test time. Various methods were proposed to tackle it under different BC setups (Ross & Bagnell, 2010; Ross et al., 2011; Sun et al., 2017; Laskey et al., 2017; Ten-

nenholtz et al., 2021; Brantley et al., 2019; Chang et al., 2021). Other issues for learning from demonstrations include non-Markovity (Mandlekar et al., 2021), discontinuity (Florence et al., 2022), randomness and noisiness (Sasaki & Yamashina, 2020; Wu et al., 2019) of the demos that lead to difficulties in neural network based policy representation and optimization and result in great compounding errors during inference. See Sec. 4.1 for detailed discussions.

**LfD as Sequence Modeling** A recent research trend in BC and offline RL is to relax the Markovian assumption of policies in the Markovian Decision Process (Howard, 1960) setup. With the widespread success of various sequence modeling models (Graves & Graves, 2012; Chung et al., 2014; Vaswani et al., 2017), model expressiveness and capacity are preferred over algorithmic sophistication. Among these, (Dasari & Gupta, 2021; Mandi et al., 2021) study one-shot imitation learning, (Lynch et al., 2020; Singh et al., 2020) explore behavior priors from demos, (Chen et al., 2021; Liu et al., 2022; Janner et al., 2021; Shafiullah et al., 2022; Ajay et al., 2022; Janner et al., 2022) examine different sequence modeling strategies for policy learning. In particular, methods based on Transformers (Vaswani et al., 2017; Brown et al., 2020) are extremely popular due to their simplicity and effectiveness. Through extensive comparison, we show the major advantages of our method compared to existing approaches.

**Hierarchical Approaches in Sequence Modeling and RL** Chain-of-Thought (Wei et al., 2022) refers to the general strategy of solving multi-step problems by decomposing them into a sequence of intermediate steps. It has recently been applied extensively in a variety of problems such as mathematical reasoning (Ling et al., 2017; Cobbe et al., 2021), program execution (Reed & De Freitas, 2015; Nye et al., 2021), commonsense or general reasoning (Rajani et al., 2019; Clark et al., 2020; Liang et al., 2021; Wei et al., 2022), and robotics (Xu et al., 2018; Zhang & Chai, 2021; Jia et al., 2022b; Gu et al., 2022; Yang et al., 2022). Similar ideas in the context of HRL can date back to Feudal RL (Dayan & Hinton, 1992) and the option framework (Sutton et al., 1999). Inspired by these approaches, ours focuses on the imitation learning setup (without reward labels or online interactions) for low-level control tasks.

**Demonstrations for Robotics Tasks** In practice, the optimality assumption of the demos is usually violated especially for robotics tasks. Demos for tasks involving low-level actuator actions primarily come in three forms: human demo captured via teleoperation (Kumar & Todorov, 2015; Vuong et al., 2021), expert demo generated by RL agents (Mu et al., 2021; Chen et al., 2022; Jia et al., 2022a), or trajectories found by planning-based methods involving heuristics, sampling and search (Gu et al., 2023; Qureshi

et al., 2019; Fishman et al., 2022). These demos are in general sub-optimal due to either human bias, imperfect RL agents, or the nature of the planners. In our experiments, we utilize planned demonstrations provided by ManiSkill2 (Gu et al., 2023). This benchmark is not saturated for IL (being adequately challenging) and has scalable sub-optimal demonstrations to bootstrap from. However, leveraging these demos to solve complex low-level control tasks in an IL setup is very challenging, as discussed in Sec. 4.1.

## 3. Preliminaries

**MDP Formulation** One of the most common ways to formulate a sequential decision-making problem is via a Markov Decision Process, or MDP (Howard, 1960), defined as a 6-tuple $\langle S, A, \mathcal{T}, \mathcal{R}, \rho_0, \gamma \rangle$, with a state space $S$, an action space $A$, a Markovian transition probability $\mathcal{T} : S \times A \to \Delta(S)$, a reward function $\mathcal{R} : S \times A \to \mathbb{R}$, an initial state distribution $\rho_0$, and a discount factor $\gamma \in [0, 1]$. An agent interacts with the environment characterized by $\mathcal{T}$ and $\mathcal{R}$ according to a policy $\pi : S \to \Delta(A)$. We denote a trajectory as $\tau_\pi$ as a sequence of $(s_0, a_0, s_1, a_0, ..., s_t, a_t)$ by taking actions according to a policy $\pi$. At each time step, the agent receives a reward signal $r_t \sim \mathcal{R}(s_t, a_t)$. The distribution of trajectories induced by $\pi$ is denoted as $P(\tau_\pi)$. The goal is to find the optimal policy $\pi^*$ that maximizes the expected return $\mathbb{E}_{\tau \sim \pi}[\sum_t \gamma^t r_t]$. Notice that, in robotics tasks and many real-world applications, the reward is at the best only sparsely given (e.g., a binary success signal) or given only after the trajectory ends (non-Markovian).

**Behavior Cloning** The most straightforward approach in IL is BC, which assumes access to pre-collected demos $D = \{(s_t, a_t)\}_{t=1}^N$ generated by expert policies and learns the optimal policy with direct supervision by minimizing the BC loss $\mathbb{E}_{(s,a) \sim D}[-\log \pi(a|s)]$ w.r.t. a mapping $\pi$. It requires the learned policy to generalize to states unseen in the demos since the distribution $P(\tau_\pi)$ will be different from the demo one $P(\tau_D)$ at test time, a challenge known as distribution shift (Ross & Bagnell, 2010). Recently, several methods, particularly those based on Transformers, are proposed to relax the Markovian assumption. Instead of $\pi(a_t|s_t)$, the policy represents $\pi(a_t|s_{t-1}, s_{t-2}, ..., s_{t-T})$ or $\pi(a_t|s_{t-1}, a_{t-1}, ..., s_{t-T}, a_{t-T})$, i.e., considers the history up to a context size $T$. This change was empirically shown to be advantageous (also justified in Sec. 4.1).

## 4. Method

In this section, we first illustrate the optimization and generalization challenges in learning from scalable yet suboptimal machine-generated demonstrations through an example. We then introduce Chain-of-Thought Predictive Control (CoTPC), which features hierarchical planning ideas

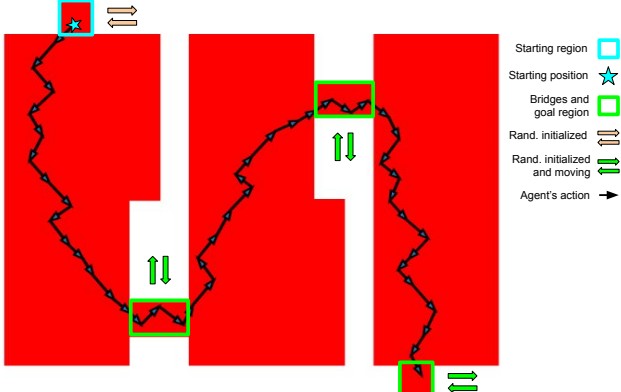

Starting region ▢
Starting position ☆
Bridges and
goal region ▢
Rand. initialized ⇉
Rand. initialized
and moving ⇉
Agent's action ➔

*Figure 2.* A moving maze where each demo trajectory is generated by a planning algorithm (with hindsight knowledge) shown as a sequence of arrows. The starting region is in the top left corner and the goal region is in the bottom right corner. Such demos are extremely scalable but at the cost of issues discussed in Sec. 4.1

to handle these challenges. Finally, we discuss several key design choices of CoTPC.

### 4.1. Moving Maze - An Illustrative Example

We present a 2D maze with a continuous action space of displacement $(\delta x, \delta y)$. As shown in Fig. 2, in this s-shaped maze, the agent starts from a location randomly initialized inside the top square region (marked in cyan) and the goal is to reach the bottom one (marked in green). Upon each environment reset, the two regions as well as the two rectangular bridges (marked in green) have their positions randomized. During the game, each of them except for the top square moves (independently) back and forth with a randomized constant speed. Once the agent lands on a moving block, the block will immediately become static. The agent cannot cross the borders of the maze (but it will not die from doing so). For simplicity, we adopt a 10-dim state observation consisting of the current location of the agent and the four marked regions.

To enable policy learning from demonstrations, we curate demo trajectories (each with a different environment random seed) by adopting a mixture of heuristics and an RRT-style planner with hindsight knowledge not available at test time (see details in the Appendix). This setup follows recent work (Gu et al., 2023) that leverages machine-generated demonstrations which are extremely scalable yet highly noisy and sub-optimal. In fact, as demonstrated in Tab. 1 & 2, a vanilla BC agent (with a multi-layer perceptron) *struggles to scale well on these demonstrations*. We summarize some common optimization and generalization challenges in learning from sub-optimal demos (especially machine-generated ones).

*Table 1.* Results for the illustrative moving maze environment. Training success rates, SRs, (%) are reported over the 200 training env. seeds. Test SRs are reported over 100 or 400 unseen seeds.

|  | VANILLA BC | D.T. | CoTPC (OURS) |
|---|---|---|---|
| MOVING MAZE | 14.5 | 76.5 | 82.0 |
| MOVING MAZE (UNSEEN) | 9.0 | 23.0 | 37.0 |

*Table 2.* With the highly noisy and sub-optimal demos, more training trajectories per vanilla BC agent can lead to a significant drop in average performance over all agents for a total of 128 trajectories. Here we show the average success rate (%).

| # TRAJECTORIES PER BC AGENT | 4 | 16 | 64 | 128 |
|---|---|---|---|---|
| MOVING MAZE | 54.7 | 38.3 | 21.1 | 16.4 |
| P&P CUBE | 33.6 | 19.5 | 7.8 | 6.3 |

**Non-Markovity** While each trajectory in the demos can be represented by a Markovian policy, the Markovian policy linearly combined from them by perfectly imitating the combined demos can suffer from a negative synergic effect if there are conflicts across demos. This is because the demos might be generated by different agents or different runs of the same algorithm. It becomes even worse when the demonstrations themselves are generated by non-Markovian agents (e.g., in Fig. 2). Instead, a non-Markovian policy is more universal and can resolve conflicts by including history as an additional context to distinguish between demos.

**Noisiness** Sometimes the demo trajectories are intrinsically noisy with divergent actions produced given the same states, e.g., with planning-based methods as in Fig. 2. This leads to increased uncertainty and variance of the cloned policies and so higher compounding errors. Note that multi-modality is a related but orthogonal issue (Shafiullah et al., 2022), i.e., when a unimodal estimate of the (continuous) action distribution leads to a significantly worse return.

**Discontinuity** For low-level control tasks, demo policies often consist of sharp value changes or topology changes (e.g., due to contact changes). Such discontinuity in the underlying state-to-action mapping leads to difficulties in learning a robust and accurate model, thus harming generalizability. A recent method (Florence et al., 2022) deals with this by an energy-based implicit model in place of an explicit one. While theoretically sound, it is shown (Shafiullah et al., 2022) to be less practical for non-Markovian implicit models, and later explicit models outperform it.

**Randomness** The actual or apparent unpredictability usually exists in sub-optimal demonstrations either because the intermediate computations of the demonstrators are not revealed in the demos (e.g., the shortest paths generated by BFS do not reveal the intermediate search process), or the

demonstrators are inherently non-deterministic (e.g., relying on rejection sampling). Such a trait makes IL less robust as the decision-making patterns from demos might be unclear, hard to learn and not generalizable (Paster et al., 2022). Procedure Cloning (Yang et al., 2022) handles a similar issue by imitating the intermediate computations. However, it assumes full knowledge of the demonstrators and it can be hard to imitate these computations (e.g., a sampling process) by neural networks.

## 4.2. Chain-of-Thought Predictive Control

To develop a scalable imitation learning algorithm for diverse and large-scale yet sub-optimal demonstrations towards the eventual advent of *decision foundation models*, we propose to incorporate the essence of HRL into a novel behavior cloning algorithm with a customized Transformer design. We leverage the hierarchical guidance provided by certain subsequences of states (or in general, frames) extracted from the demonstrations. Our model learns to dynamically predict such hierarchical structures as coherent long-term action guidance, which enables high-level planning capabilities and substantially eases complex low-level control tasks.

**Key States as Chain-of-Thought (CoT)**   We observe that many low-level control tasks (e.g., object manipulations) naturally consist of sequences of subgoals. And there exist key states, each of which marks the completion of a subgoal. We name them the chain-of-thought (CoT), as they represent the global structure of a decision-making process and can provide coherent, succinct yet long-term behavior guidance. For instance, in the moving maze in Fig. 2, the two bridges naturally divide the task into three subgoals. Given a demo trajectory, we extract three key states - when the agent first reaches the bridges and the goal region. Formally, for each trajectory $\tau \in D$, we define CoT as an ordered set of states $F_{cot}(\tau) = \{s_t | s_t \in \tau\} = \{s_k^{cot}\}$. We describe the underlying principles to determine them shortly in Sec. 4.3.

**Coupled CoT and Behavior Modeling**   We find that compared to the demos' overall high variance and noises, the CoT admits much fewer variations and shares a generalizable pattern among demos (also explained in Sec. 4.3). Hence, action predictions conditioned on the CoT (the key states) ease model optimization with reduced variations of inputs (the consistent future cues become part of the inputs). Moreover, the impressive sequential reasoning capability of Transformers (Brown et al., 2020) enables our model to acquire (even from demonstrations of heavy noises) the capability of long-term planning via CoT prediction. As the action and CoT predictions are complementary, we propose to couple them in a shared Transformer architecture.

**Dynamic CoT Prediction with Transformer**   We adopt a customized version of GPT (Brown et al., 2020) to enable key state predictions alongside action predictions both in a contextual manner. We insert $K$ different learnable tokens at the beginning of the state and action context history. We set $K$ as the number of key states, which is task-specific (e.g., $K = 3$ for peg insertion). These serve as prompt tokens similar to CoOp (Zhou et al., 2022) and are learned to predict the corresponding key states given the current trajectory context. Instead of using a causal attention mask, we design a *hybrid* masking regime, where normal state or action tokens can attend to those in the past (standard causal mask) and so *always* see the key state tokens for action predictions (achieved with an action decoder). The key state prompt tokens, however, are all-to-all and can observe *all* action and state tokens in the current context window (i.e., until the current time step $t$). Formally, given a context size of $T$ with a sampled trajectory subsequence up to some timestep $t$, i.e.,

$$\tau_T(t) = \{s_{t-(T-1)}, a_{t-(T-1)}, ..., s_{t-1}, a_{t-1}, s_t\}$$

we apply the hybridly masked multi-head attention layer, denoted $\text{MHA}_{hmask}[\cdot]$, to features of $\tau_T(t)$. Namely,

$$h_j(\tau_T(t)) = \text{MHA}_{hmask}[F_{enc}(\tau_T(t))], \quad j = 1$$
$$h_j(\tau_T(t)) = \text{MHA}_{hmask}[h_{j-1}(\tau_T(t))], \quad j > 1$$

where $F_{enc}$ encodes each action token and state token by encoder $f_a(\cdot)$ and $f_s(\cdot)$, respectively (no encoder for the prompt tokens). Here we omit the position embeddings and the additional operations between the attention layers in standard Transformers. We use two decoders $g_a(\cdot)$ and $g_{cot}(\cdot)$ to predict the current action $\hat{a}_t$ and the current key state predictions $\{\hat{s}_{k,t}^{cot}\}$, respectively, i.e.,

$$\hat{a}_t = g_a(h_J(\tau_T(t))[-1])$$
$$\hat{s}_{k,t}^{cot} = g_{cot}(h_I(\tau_T(t))[k]), \ k \in \{0, ..., K-1\}$$

where $J$ is the index of the last attention layer and $[k]$ selects the $k$-th element as in Python (each key state token is used to predict a different key state for better performance). Note that $I \in \{1, ..., J\}$, i.e., the key state decoder takes the outputs of the $I$-th attention layer, where $I$ is a hyperparameter. We find $I = 1$ to work well for most tasks. To summarize, our model *jointly generates actions and key states at each step* by conditioning on the trajectory history. The key states are *dynamically* updated during interactions, resembling closed-loop control. This design can be immediately justified in the case of dynamic environments such as the moving maze or tasks that require dynamic controls. We will discuss the key design choices in more detail shortly.

**Training Objective**   The overall training pipeline is illustrated in Fig. 1. The model is trained with behavior cloning

loss as well as the auxiliary key state prediction loss $\mathcal{L}_{cot}$ based on MSE (weighted by a coefficient $\lambda$), which yields the overall training objective:

$$\mathcal{L}_{total} = \mathop{\mathbb{E}}_{(s_t, a_t) \in D} \mathcal{L}_{bc}(\hat{a}_t, a_t) +$$

$$\frac{\lambda}{K} \sum_{k=0}^{K-1} \mathop{\mathbb{E}}_{\tau \in D} \frac{1}{|\tau|} \sum_{t=0}^{|\tau|-1} \mathcal{L}_{cot}(\hat{s}_{k,t}^{cot}, s_k^{cot})$$

See more implementation details in the Appendix.

### 4.3. Principles of Chain-of-Thought Selection

As discussed previously, the underlying principle for choosing key states is to find a shared common pattern across demos that can improve the optimization and generalization of BC. Specifically, for each trajectory $\tau \in D$, we aim to find an ordered set of states $F_{cot}(\tau) = \{s_t | s_t \in \tau\} = \{s_k^{cot}\}$ that can satisfy the following principles:

- The reaching of key states should indicate the completion of the subgoals (and eventually of the task).

- The key states should admit fewer variations than the average of other states in the trajectories.

- Functional correspondence (with a fixed order) exists between key states of different trajectories.

We find that locating key states with these conditions is easy for object manipulation tasks, as they are often inherently hierarchical with multiple sub-stages. Roughly speaking, if all trajectories for a contact-rich manipulation task share the same sequence of sub-stages with a fixed order, then the boundaries of adjacent sub-stages constitute the key states, and the patterns are shared across trajectories. For example, in peg insertion, the robotic gripper goes through the grasp stage to reach the peg (the first key state occurs when it first grasps the peg), then the align stage that ends up aligning the peg with the hole (another key state), and eventually, the insert stage, the end of which (the last key state) marks the task completion. We can easily derive heuristic rules to automatically recognize such key states from demos in simulations with the help of privileged information such as contact feedback and object poses.

In our experiments and ablation studies (Sec. 5.6), we empirically justify our principles of key states selection and compare our design with several alternatives. To summarize, our design reflects a trade-off between leveraging the hierarchy of the tasks and keeping CoT prediction tractable.

### 4.4. Design Choices in Chain-of-Thought Predictions

At test time, our model predicts key states (CoT) along with actions. Here we will discuss several relevant design options. Please also refer to the ablation studies in Sec. 5.6.

**Static vs. Dynamic CoT Prediction** Static prediction refers to predicting the key states only once at the very beginning of each trajectory. Dynamic prediction, on the other hand, keeps updating its key state predictions along the way to task completion. The difference between the two is somewhat similar to open-loop vs closed-loop control (also see Sec. 4.2). We find that the dynamic version is significantly more robust as the model adjusts its prediction of both key states and actions on-the-fly. In the case of human disturbance (e.g., pushing an object during task execution) or dynamic environments (such as the moving maze), the dynamic version obviously becomes a necessity.

**Contextual vs. Markovian CoT Prediction** A Markovian CoT prediction strategy predicts the key states at each step by conditioning only the current state. A non-Markovian one considers the context history of states and actions (up to a maximum number of steps) for both key state and action prediction. We empirically find that this strategy outperforms the Markovian one by a large margin. When combined with Transformer-based policies, contextual and dynamic CoT prediction can be achieved gracefully, as described in Sec. 4.2.

**CoT Predictions in the Latent Space** We further observe that predicting key states in the latent space (i.e., the actual key states are further predicted via a non-linear two-layer MLP decoder) outperforms the alternative (predicting them more explicitly such that we only use a linear decoder). This potentially leads to a better representation of the predicted CoT than the manually designed state observation, as the action prediction loss (the BC loss) also encourages the CoT prediction process to provide better behavior guidance.

**CoT Prediction vs. Subgoal Prediction** While inspired by HRL, CoTPC differs from typical two-stage subgoal-conditioned policies. Instead of only the immediate next subgoal, we predict an entire chain of them all at once and at each inference step. This has several major advantages. Existing subgoal-conditioned policies mostly predict the immediate subgoals either sporadically or fully dynamically. For sporadic ones, CoTPC can better handle dynamic environments (or those requiring dynamic control) with its dynamically adjusted subgoal-level plans. For fully dynamic ones, these policies have to approximate a very discontinuous function (sharp change of predicted subgoal near the sub-task boundaries), which leads to robustness issues. Alternatively, CoTPC predicts subgoals altogether in multiple slots so that conditioning action predictions on a different subgoal is simply *attending to different key state tokens*. Moreover, only predicting the immediate next subgoal means predicting subgoals sequentially (i.e., auto-regressively), which is more vulnerable to compounding errors than our joint prediction strategy (Qi et al., 2020).

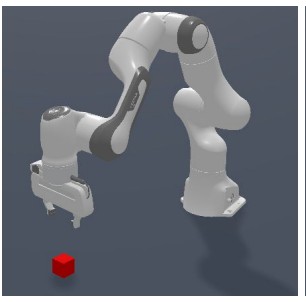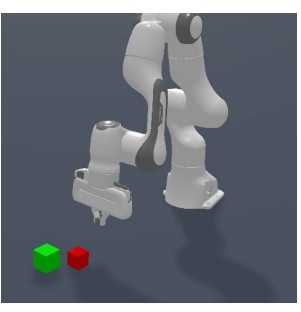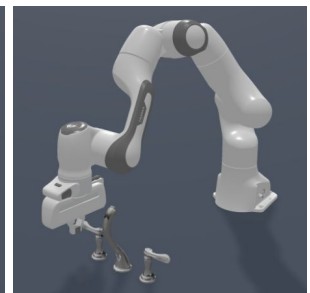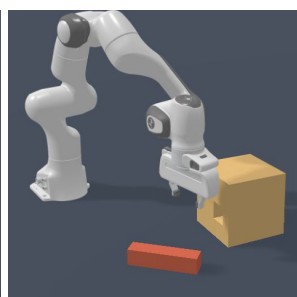

*Figure 3.* We use four challenging contact-rich object manipulation tasks from ManiSkill2 (Gu et al., 2023) to evaluate our method. Namely, P&P Cube (pick and place cube), Turn Faucet, Stack Cube and Peg Insertion Side. We use state-space observations in our experiments while the images are rendered for illustration (the goal location varies across seeds but is not visualized for P&P Cube).

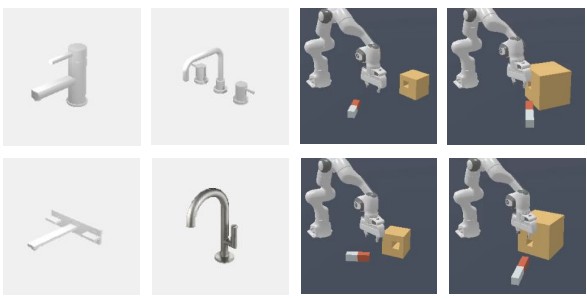

*Figure 4.* Sampled geometric variations for turn faucet and peg insertion. The sizes of peg / box and the relative locations of the hole vary across different environment seeds.

# 5. Experiments

In this section, we first introduce the tasks and the metrics used to evaluate our method, discuss the baselines we compare with, and then present the main results. Finally, we show ablation studies that justify our design choices.

## 5.1. Tasks and Environments

We focus on low-level object manipulation tasks in an imitation learning setup. We choose ManiSkill2 (Gu et al., 2023), an extension of ManiSkill (Mu et al., 2021) as the testbed, which features a variety of object manipulation tasks in environments with *realistic physical simulation* (including fully dynamic grasping motions). We choose four tasks as illustrated in Fig. 3. Namely, P&P Cube which asks the agent to pick up a cube, move it at a specified goal location, and keep skill for a short while; Stack Cube for picking up a cube, placing it on top of another, and the gripper leaving the stack; Turn Faucet for turning on different faucets; and lastly, Peg Insertion Side for inserting a cuboid-shaped peg sideways into a hole in a box of different geometries and sizes. We notice that existing benchmarks are either saturated for imitation learning methods, e.g., DMControl

(Tassa et al., 2018) and D4RL (Fu et al., 2020), or lack demo data, such as MineDojo (Fan et al., 2022). The tasks we choose lie in between, which is challenging and has sub-optimal demo data to bootstrap from.

The challenges of low-level manipulation tasks we selected come from several aspects. Besides noises injected into the initial robot pose, all tasks have all object poses randomized (displacement around 0.3m and $360°$ rotation) upon environment reset. Moreover, both turn faucet and peg insertion have large variations on the geometries and sizes of the target objects (see Fig. 4). They are particularly challenging also because the faucets are mostly pushed rather than grasped during manipulation (under-actuated control), the holes have 3mm clearance, and it requires at least half of the peg to be pushed sideway into the holes (harder than similar tasks in other benchmarks).

As an early exploration of applying chain-of-thought towards low-level control, we mainly study state-based policy learning to avoid the confounding challenges of visual perception learning (see details of observation space in the Appendix and the ManiSkill2 paper). We use an 8 DoF continuous action space for delta joint position control, which is native to the demonstrations generated by many planning-based methods.

## 5.2. Demonstrations Data

The complexity of our imitation learning tasks also lies in the highly noisy and sub-optimal demonstrations. The demos provided by ManiSkill2 are generated by a mixture of multi-stage motion-planing and heuristics-based policies (with the help of privileged information in simulations) in the form of state-action sequences without reward labels. For wide applicability, we do not assume detailed knowledge of the demonstrators (which is not available from ManiSkill2). We randomly sample 500 training trajectories for each task (50 trajectories per faucet model over 10 faucets in turn facet). We also double the demo data for peg inser-

*Table 3.* Results for tasks in ManiSkill2 on seen environment seeds. We compare ours with several recent baselines (adapted to the BC setup) to show how it tackles the optimization challenges. Task SR denotes the task success rate. For the challenging peg insertion task, we also report intermediate success rates other than Insert SR. All metrics are reported in percentage (%) with the best ones **bolded**.

| | P&P CUBE | STACK CUBE | TURN FAUCET | PEG INSERTION | | | MEAN (%) |
|---|---|---|---|---|---|---|---|
| | TASK SR | TASK SR | TASK SR | GRASP SR | ALIGN SR | INSERT SR | TASK SR |
| VANILLA BC | 3.8 | 0.0 | 15.6 | 58.8 | 1.2 | 0.0 | 4.9 |
| BEHAVIOR TRANSFORMER | 23.6 | 1.6 | 16.0 | 90.0 | 17.0 | 0.8 | 10.5 |
| DECISION DIFFUSER | 11.8 | 0.6 | 53.6 | 86.8 | 9.2 | 0.6 | 16.7 |
| MASKDP + GT KEY STATES | 54.7 | 7.8 | 28.8 | 62.6 | 5.8 | 0.0 | 22.8 |
| DECISION TRANSFORMER | 65.4 | 13.0 | 39.4 | 97.8 | 41.8 | 5.6 | 30.9 |
| COTPC (OURS) | **75.2** | **58.8** | **56.4** | **99.6** | **98.2** | **52.8** | **60.8** |

*Table 4.* Results (%) on unseen environment seeds to evaluate generalizable policy learning. We only show DT here as we find it to have the best generalization performance among the baselines. Metrics involving 0-shot generalization (over unseen geometries) are *italicized*. Results obtained from models trained with demo trajectories of doubled sizes are marked with $\times$. Best results are **bolded**.

| | P&P CUBE (UNSEEN) | STACK CUBE (UNSEEN) | TURN FAUCET (UNSEEN & 0-SHOT) | | PEG INSERTION (0-SHOT) | | | |
|---|---|---|---|---|---|---|---|---|
| | TASK SR | TASK SR | TASK SR | *Task SR* | *Grasp SR* | *Align SR* | *Insert SR* | *Insert SR$^{\times}$* |
| DECISION TRANSFORMER | 50.0 | 7.0 | 32.0 | 9.0 | 92.3 | 21.8 | 2.0 | 3.5 |
| COTPC (OURS) | **70.0** | **46.0** | **57.0** | **31.0** | **95.3** | **72.3** | **16.8** | **38.0** |

tion since it requires generalization over unseen geometries similar to a zero-shot transfer setup. Even for the relatively simple pick cube task, we find vanilla BC agents struggle to scale well regarding sub-optimal demonstrations. Over a total of 128 demo trajectories evenly assigned to $S$ agents, we find that the smaller $S$ is (meaning each agent learns from more demos), the lower the average training performance becomes, as reported in Tab. 2.

### 5.3. Training and Evaluation Protocols

For each task, we train all methods in the behavior cloning setup on the same set of demo trajectories, where each trajectory is generated with a different environment seed (variation). At test time, we evaluate in the simulated environment using both the seen and unseen seeds to investigate how well our model tackles the optimization and the generalization challenges. For p&p cube, stack cube and turn faucet, we use 100 unseen seeds. For turn faucet, we further use 400 seeds over a held-out set of 4 unseen faucet models (a zero-shot transfer setup). For peg insertion, we use 400 unseen seeds which yield unseen geometries of both the peg and the hole (also a zero-shot transfer setup). We use success rate (SR) as the major metric. Our tasks and the LfD setup are so challenging that most existing state-of-the-art methods struggle at the optimization phase. We further provide some intermediate success rates as additional metrics for a more informative comparison. See the Appendix for detailed descriptions of these metrics.

### 5.4. Baselines

We first train a vanilla BC policy with a three-layer MLP and find that it performs extremely poorly on all tasks. We then compare our method with several popular non-Markovian baselines that achieve state-of-the-art performance. Namely, Decision Transformer (DT) (Chen et al., 2021), Behavior Transformer (BeT) (Shafiullah et al., 2022), MaskDP (Liu et al., 2022) and Decision Diffuser (DD) (Ajay et al., 2022).

DT is originally proposed for offline RL that applies sequence modeling to demonstrations with densely labeled rewards. We adapt DT for the BC setup by ignoring the reward tokens. We implement our proposed CoTPC on top of DT by sharing the same configurations for the Transformer backbone. BeT is proposed to handle multi-modalities issues in BC and is claimed to be able to model arbitrary multi-modal state-action distributions via an action space discretization strategy. As multi-modalities is a relevant yet orthogonal issue, we do not find it to work well in our tasks (which are more challenging than the ones used in their original paper), potentially because of the lack of high precision due to its discretization process. MaskDP, on the other hand, leverages masked auto-encoding (He et al., 2022), a bi-directional sequence modeling technique to improve the generalization of BC. It features action predictions conditioned on goal states (similar to key states), with a limitation that it requires ground truth key states with their precise time steps provided to the model at test time. Ours, instead, predict actions and key states jointly. DD explores the dif-

*Table 5.* Results (%) on seen environment seeds for various ablation studies over the key state selection and the key state prediction strategies. See Sec. 5.6 for details.

| | STACK CUBE | PEG INSERTION | | |
| --- | --- | --- | --- | --- |
| | TASK SR | GRASP | ALIGN | INSERT |
| PRED LAST STATE ONLY | 12.0 | 96.8 | 65.6 | 9.6 |
| PRED ALL STATES | 30.0 | 92.4 | 70.2 | 18.0 |
| PRED ALT KEY STATES | 17.0 | 80.0 | 44.8 | 5.4 |
| PRED MARKOVIAN | 22.2 | 78.8 | 41.4 | 6.4 |
| PRED LINEAR DEC | 21.6 | 97.4 | 51.6 | 11.2 |
| PRED W/ ALT DECODER | 33.0 | 96.0 | 65.8 | 12.8 |
| CoTPC | 58.8 | 99.6 | 98.2 | 52.8 |

*Table 6.* Ablation studies that justify why CoTPC predict the entire CoT instead of only the immediate next subgoal (as in common HRL methods). All results reported as the average task success rate (%). * means zero-shot.

| | STACK CUBE | | PEG INSERTION | |
| --- | --- | --- | --- | --- |
| | SEEN | UNSEEN | SEEN | UNSEEN* |
| CoTPC (NEXT SG) | 47.6 | 22.0 | 29.6 | 9.0 |
| CoTPC (CoT) | 58.8 | 46.0 | 52.8 | 16.8 |

fusion model (Ho et al., 2020) for policy learning by first acquiring a trajectory prediction model and then predicting actions with an inverse dynamics model.

### 5.5. Main Results

**Outstanding Performance by CoTPC** We find all the strong baselines (even the MaskDP with GT key states) are significantly outperformed by ours. Results are reported in Tab. 3 & 4, which shows the clear advantages of CoTPC from both the optimization and the generalization perspective. Most existing methods struggle with optimization, let alone generalization.

**CoTPC is Scalable** To further examine whether our proposed model can scale well regarding the sub-optimal demonstrations, we perform an evaluation with a doubled size of training trajectories. We report the results in Tab. 4, which suggests that CoTPC is able to substantially benefit from large-scale yet noisy demonstrations. We believe it is a promising step towards decision foundation models.

### 5.6. Ablation Studies

We perform two sets of ablation studies to justify our design choices, with results summarized in Tab. 5 & 6.

**Different Rules for Key States** We perform ablation studies on the key state selection strategy. In the first variant, we only include the last key state (i.e., the very last state of each demo trajectory) in the auxiliary training loss, denoted

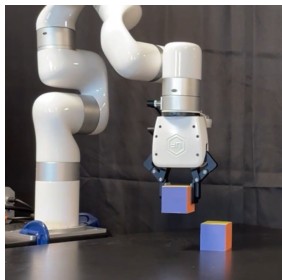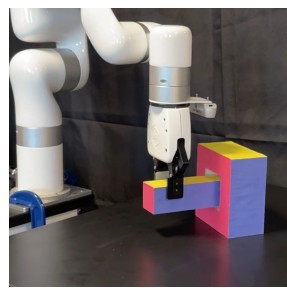

*Figure 5.* The setup in our real-world experiments for stack cube and peg insertion. As an early examination, we increase the clearance for peg insertion from 3mm (sim) to 10mm (real).

as "Pred last state only". This variant performs worse than using all the key states since it does not fully leverage the hierarchical structure of the low-level control tasks. In the second variant, we include states other than our selected key states. In fact, we ask the model to predict all states given the previous state-action pairs and the current state, similar to learning the dynamic model (see implementation details in the Appendix). The extra states in this variant lead to noises during training as most states admit large variances; the resulting model generalizes worse since the guidance from key states is weakened. We denote this variant as "Pred all states". The last variant uses an alternative set of rules to select the key states, which produce states not at the boundaries between stages but rather in the middle of each stage, which are usually of high variations. We denote this as "Pred alt key states".

**Different Key State Prediction Strategies** We first justify the use of contextual and dynamic key state predictions. As discussed in Sec. 4.2, we find that with a Transformer architecture, the contextual (predicting conditioned on the history) and dynamic (predicting on-the-fly when carrying out the predicted actions) prediction strategies can be gracefully combined. Alternatively, we use an MLP to predict the key states given only the current state $s_t$ as the inputs. This variant is denoted "Pred Markovian" since it does not use the whole context history for key state predictions. In the next variant, we empirically show the advantage of key state prediction in the latent space (discussed in Sec. 4.4). We use a linear layer as the key state decoder and denote it as "Pred Linear Dec" In another variant, we use a three-layer MLP in place of a two-layer one for the key state decoder. We denote it as "Pred w/ alt decoder". In the last variant, we make CoTPC only predict the next key state at each time with only one key state prompt token. This is reminiscent of common HRL methods that perform subgoal generation and adopt a two-stage subgoal-condition policy. The results for this variant are reported in Tab. 6 instead.

See further implementation details for these ablation studies

in the Appendix.

## 5.7. Sim-to-real Transfer

Although we mainly study CoTPC with state observations, we examine the plausibilities of the sim-to-real transfer on stack cube and peg insertion. Our real-world experiment setup is illustrated in Fig. 5. With an off-the-shelf pose estimation framework such as PVNet (Peng et al., 2019), we can achieve reasonable performance using the policy learned purely in simulations. We provide visualization of task executions in our project page and defer the details to the Appendix.

## 6. Discussions, Limitations and Future Work

**Comparison with Procedure Cloning** Procedure cloning (PC) (Yang et al., 2022) is an extension of BC by imitating the intermediate computations of the demonstrators. It requires full knowledge of how the demo trajectories are generated including the usually hidden computations (such as the status of each node at each step in BFS, one of their examples). CoTPC does not require such knowledge, as the key states are part of the results, not intermediate computations. To apply PC for demonstrations provided by, e.g., a sampling-based planner, it needs to imitate the sampling process (which is hard) by accessing the potentially large amounts of intermediate sampled paths not ended up in the demos. In general, machine-generated demonstrations can be crowd-sourced and the demonstrators are viewed as black boxes, making this a limitation of PC. As a side note, ManiSkill2 provides demonstrations with high-level descriptions but not the actual demo policies.

**Comparison with Subgoal-conditioned Policies** Unlike many existing work dealing with "long-horizon" tasks, e.g., SayCan (Ahn et al., 2022) and methods tackling ALFRED (Shridhar et al., 2020), which assume that low-level control is solved, we instead focus on solving low-level control with hierarchical information such as key states. Many methods from HRL or hierarchical IL adopt a two-stage subgoal-condition policy. CoTPC differs from them in a non-trivial aspect. Instead of only the immediate next subgoal, CoTPC predicts an entire chain of them all at once and at each inference step, which has several advantages as explained and empirically verified in Sec. 4.4.

**Comparison with RT-1** Robotic Transformer-1 (RT-1) (Brohan et al., 2022) is a concurrent work that also directly models low-level control actions with a Transformer. It benefits from the sheer scale of real-world robot demonstration data pre-collected over 17 months and the tokenization of both visual inputs (RGB images) and low-level actions.

While RT-1 shows great promise in developing decision foundation models for robotics, it adopts the conventional auto-regressive Transformer without explicitly leveraging the structural knowledge presented in low-level control tasks. Our work, CoTPC, is an early exploration in this direction and we believe it will inspire the future designs of generally applicable models for robotics tasks. Another difference is that since RT-1 discretizes the action space, it might suffer from degraded performance for tasks that require high precision (such as peg insertion).

**Limitations and Future Work** CoTPC requires prior knowledge of key states that admit reduced variance and learnable patterns shared across different trajectories. While such information is generally available for low-level manipulation tasks, we plan to further leverage the compositionality of human languages and the pre-trained LLMs to improve the key state selection process. We also plan to extend our work to high-dimensional visual inputs that can be more easily transferred to real-world robots. An integrated modeling approach involving both visual inputs, human language and low-level actions in the style of PaLM-E (Driess et al., 2023) is also a potential extension. We also believe that CoTPC can be extended in a multi-task learning setup with policy learning from diverse demos across different low-level control tasks.

## 7. Conclusions and Future Work

In this work, we propose CoTPC for learning generalizable policies from scalable but sub-optimal demonstrations. CoTPC leverages hierarchical structures in low-level control tasks (e.g., object manipulations) and, at each step, predicts actions jointly with the key states (the Chain-of-Thought) that provide structured long-term action plans. With the guidance of patterns presented in the CoT that are shared across different demos, CoTPC produces generalizable policies that significantly outperform existing methods on challenging low-level contact-rich tasks.

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

# A. Various Details for the Four Tasks Used In Our Experiments

## A.1. State space

In our experiments, we use state space observation to avoid adding the additional challenges of visual-based policy learning that might be confounding to our studies of BC. The state observations for the four tasks have dimensions ranging from 43-d (`TurnFaucet`) to 55-d (`PegInserstionSide`). The original state observation dimension for `TurnFaucet` is 40. However, as there are different faucet models (with largely varied geometries) used across different environment variations, the original state space misses key information that can be used to distinguish between different faucet models. The visual observation space does not have this issue, though. We, therefore, append an extra 3-d vector for the pose of the faucet link by replaying the demos. As a result, the agent can learn to turn different faucet models jointly.

## A.2. Demonstrations

The demonstrations provided by ManiSkill2 (generated by TAMP solvers, see their original paper for details) contain 1000 trajectories for each of the four tasks except for `TurnFaucet`, where a varied number of trajectories are provided for different faucet models. We randomly choose 10 faucet models whose demos are generated by TAMP solvers. We then perform stratified sampling to choose 500 trajectories from `TurnFaucet` and 500 trajectories randomly sampled for the other three tasks. For both training and testing, we use the same set of demos (each trajectory has an average length of around 200). During testing, we set the maximum steps allowed as 150, 250, 200, 250 for P&P Cube, Turn Faucet, Stack Cube and Peg Insertion Side, respectively.

## A.3. Key state selection

To generate parsing rules to label key states for each demonstration, one can resort to human knowledge by simply asking ChatGPT "What are the major steps to XXX?", where XXX is the high-level description of the four tasks. We then write simple heuristics programs in the SAPIEN simulated environment (Xiang et al., 2020) (where ManiSkill2 is hosted) to automatically find the key states. Specifically, for P&P Cube, there are two key states: the first state when the cube is grasped by the robotic gripper and the finishing state (the last state) in a demo trajectory when the cube is placed in the target location. For Turn Faucet, there are two key states: the first state when the gripper has made a contact with the handle of the faucet and the finishing state (the last state) in a demo trajectory when the faucet is turned on. For Stack Cube, there are three key states: the first state the cube is grasped by the robotic gripper, the last state the gripper is still grasping the cube while the cube is stacked on top of the other, and the finishing state (the last state) in a demo trajectory. Note that to complete the Stack Cube task, the gripper needs to leave both cubes aligned vertically and static. For Peg Insertion Side, there are three key states: the first state when the peg is grasped, the first state the peg's angle is aligned with the hole, and the finishing state (the last state) in a demo trajectory where the peg is inserted (at least half of it) into the hole.

## A.4. Evaluation metrics

While the success conditions are well-defined by the authors of ManiSkill2 for the four tasks, we also derive intermediate success conditions to better evaluate the performance of different models. Specifically, for Peg Insertion Side, "Grasp SR" is defined as whether at any time step, the peg has been grasped by the gripper successfully, "Align SR" is defined as whether at any time step, the peg's angle has been adjusted to be aligned with the hole.

# B. More Details for the Baselines and Our Proposed Method

**MLP baseline (vanilla BC)**: It is a three-layer MLP with a hidden size of 256 and ReLU non-linearity. We train it with a constant learning rate of $1e-3$ with Adam optimizer with a batch size of 32 for 150K iterations (training longer leads to over-fitting even with tuned L2 regularization).

**Decision Transformer:** We use the exact same configurations, for training and for the Transformer architecture (excluding the parts regarding key state tokens), in DT as our method. In our proposed model, the action decoder and key state decoder are both 2-layer MLPs of one hidden layer of size 256 and with ReLU non-linearity. We train it with a learning rate of $5e-4$ (with a short warm-up period and cosine decay schedule to $5e-5$) with the Adam optimizer with a batch size of 256 for 1.8M iterations. We use a weight decay of 0.001. We use a context size of 60 for all tasks. We find that our model, as well as DT that shares the same Transformer architecture and BC loss, can train much longer than others without worrying

about overfitting.

**Behavior Transformer:** We started with the configuration used for the Franka Kitchen task in the original paper. We changed the number of bins in K-Means to 1024 (we find that for our tasks, a smaller number of bins works worse) and changed the context size to 60 (in line with the other transformer-based models). The Transformer backbone has approximately the same number of parameters ($\sim$1M) as ours and DT. We train the model for around 50k iterations (we find that training longer leads to over-fitting easily for BeT, potentially because of its discretization strategy and the limited demos used for BC).

**Decision Diffuser**: We use the reference implementation provided by the authors and make the following changes in the diffusion model: 100 diffusion steps, 60 context size, and 4 horizon length (in our experiments we found that longer performs worse). The diffusion and inverse-dynamics models have $\sim$1.6M parameters in total. Since DD works on fixed sequence lengths, we pad the start and end states during training and only the start states during inference.

Some existing methods (Florence et al., 2022; Yang et al., 2022) are not included because either they were compared unfavorably with the baselines or they assume access to information impractical in our tasks. For each baseline (and our method), we report the evaluation results using the best checkpoint along the training (since we are evaluating the same set of environment variations used to generate the demo, this is essentially validation performance).

## C. Details for the Ablation Studies

For the variant denoted as "Pred all states", we ask the model to predict all states given the previous state-action pairs and the current state, similar to learning the dynamic model. Specifically, in the Transformer backbone, we apply a next-state decoder (two-layer MLP similar to the action decoder) for each state token and train the model to predict the very next state (via an auxiliary MSE loss). In this variant, there are no query tokens for the key states as the state predictions are performed alternatively. In the variant denoted as "Pred Markovian", the key state decoder always takes the last state in the current context history as the only input and predicts the key states in a coupled manner. We use a three-layer MLP with a hidden size of 256. As the decoder is unaware of the states and actions occurring before the current one in the context buffer, it is essentially a Markovian predictor.

## D. More Implementation Details for Model Training

We adopt the minGPT implementation and use the same set of hyperparameters for all tasks (a feature embedding size of 128 and 4 masked multi-head attention layers, each with 8 attention heads, totalling around 1M learnable parameters). We use no positional embeddings for key state tokens and use learnable positional embedding for the state and action tokens similar to DT (Chen et al., 2021). The action decoder and key state decoder are both 2-layer MLPs. We use a coefficient $\lambda = 0.1$ for the auxiliary MSE loss. While any hidden layer can be utilized for key state predictions, we find the resulting performance varies (some can lead to training instabilities) and only use the first hidden layer for all tasks (making it task-specific does increase the performance, though). During training, for efficiency, we apply random masking to the action and state tokens so that the key states tokens attend to a history of varied length (from the first step to some $t$ in the context history). At test time, while the key state decoder is unused, the action predictor attends to all tokens in the past (including the key states tokens) and the key state tokens (as is all-to-all) attend to all tokens up to the current time $t$. We train CoTPC similar to the DT baseline with a learning rate of $5e - 4$ (with a short warm-up period and cosine decay schedule to $5e - 5$) with the Adam optimizer with a batch size of 256 for 1.8M iterations. We use a weight decay of $0.001$.