# OpenReview forum: "Chain-of-Thought Predictive Control"
_ICLR.cc/2023/Workshop/RRL — RRL 2023 Poster_

### Official Review · Reviewer_3MKn · 2023-02-20
**Interesting Idea**

**Rating:** 3
**Confidence:** 4

**Review:**

The paper presents an approach for learning policies for long-horizon tasks (e.g., peg insertion) that relies on predicting "key states" along the trajectory. The main idea is that key states are a) easily predictable and b) useful for conditioning action predictions. The results show improvements over baseline methods on four tasks: pick cube, turn faucet, stack cube, and peg insertion. The approach uses ChatGPT to describe the steps for solving each task and then the authors design heuristic rules for each step to identify "key states" in expert demonstrations, which are used for learning (via an auxiliary training objective). The paper is clearly written.

The paper does not use RL and the use of prior knowledge (from ChatGPT) is limited. However, the approach should still foster interesting conversations at the workshop. Specifically, the high-level idea of using an LLM to decompose tasks and then using that decomposition for learning seems highly relevant.

The authors could consider other ways in which LLMs could be used in the propose approach. For example, can the LLM produce the heuristics rules for identifying the keys states after the steps for solving the task have been listed? Or can an LLM be used to propose methods for identifying key states without the need for privileged simulator information? Such techniques may help scale the proposed approach to more tasks and more domains.

---

### Official Review · Reviewer_5izj · 2023-03-01
**The paper proposes a Transformer based method for Imitation Learning which shows very good performance on complex Control problems. The experimental results are very good but there are some caveats with the proposed method and the baselines used.**

**Rating:** 3
**Confidence:** 3

**Review:**

This paper proposes a Transformer-based imitation learning method for solving complex robotic control tasks. The paper is clearly written and understandable. There are several illustrations that can help with understanding the proposed method. The proposed method named CoTPC outperforms several other BC methods by significant margins. It manages to do so by encoding knowledge about the multiple intermediate subgoals that need to be reached so that the ultimate task can be solved. Below are the strengths and weaknesses of the paper:

Strengths:

1)  The Key State concept for dealing with complex continuous control problems which include many intermediate subproblems is interesting and allows for knowledge about the multi-step nature of the problem to be incorporated into a BC method.

2) The experimental results are very good compared to the other methods and the other baselines are recent successful BC methods.

3) The examples and the illustrations that are provided help with demonstrating the necessity for using key states in these problems.


Weaknesses:

1) One of the methods used as a baseline for the experimental comparisons is the Decision Transformer (DT) approach which is an offline RL method, not a BC one. It requires return inputs to properly function. The DT implementation used in this paper ignores reward tokens and this can possibly hurt performance significantly. Including DT as a baseline might be unfair.

2) The addition of other Control benchmarks would have been helpful to paint a clearer picture on the differences in performance.

3) It is stated that ChatGPT is used for automating the process of producing the subgoals needed for each task. This seems completely unnecessary as there are only 4 problems each with very few intermediate steps. However, this is a very minor detail and does not change the assessment overall.


Question:

Could the key state approach be problematic in some environments? The states are continuous-valued and there could be possibly infinite different choices for a key state at each intermediate step.

The contributions are marginally significant or novel as it is not a major overhaul of transformer-based BC (or offline RL) methods.

Overall, I am leaning towards acceptance.